# Acoustic Neurinoma with Synchronous Ipsilateral Cerebellopontine Angle Lipoma: A Case Report and Review of the Literature

**DOI:** 10.3390/diagnostics12010120

**Published:** 2022-01-05

**Authors:** Takahiro Kanaya, Yasuo Murai, Kanako Yui, Shun Sato, Akio Morita

**Affiliations:** 1Department of Neurosurgery, Dokkyo Medical University Saitama Medical Center, 2-1-50 Minamikoshigaya, Koshigaya 343-8555, Saitama, Japan; t-kanaya@nms.ac.jp; 2Department of Neurological Surgery, Nippon Medical School, 1-1-5 Sendagi, Bunkyo-ku 113-8603, Tokyo, Japan; y-kanako@nms.ac.jp (K.Y.); s3049@nms.ac.jp (S.S.); amor-tky@umin.ac.jp (A.M.)

**Keywords:** cerebellopontine angle, lipoma, synchronous tumor, neurinoma

## Abstract

Lipomas of the cerebellopontine angle (CPA) and internal auditory canal (IAC) are relatively rare tumors. Acoustic neurinoma is the most common tumor in this location, which often causes hearing loss, vertigo, and tinnitus. Occasionally, this tumor compresses the brainstem, prompting surgical resection. Lipomas in this area may cause symptoms similar to neurinoma. However, they are not considered for surgical treatment because their removal may result in several additional deficits. Conservative therapy and repeated magnetic resonance imaging examinations for CPA/IAC lipomas are standard measures for preserving cranial nerve function. Herein, we report a case of acoustic neurinoma and CPA lipoma occurring in close proximity to each other ipsilaterally. The main symptom was hearing loss without facial nerve paralysis. Therefore, facial nerve injury had to be avoided. Considering the anatomical relationships among the tumors, cranial nerves, and CPA/IAC lipoma, we performed total surgical removal of the acoustic neurinoma. We intentionally left the lipoma untreated, which enabled facial nerve preservation. This report may be a useful reference for the differential diagnosis of similar cases in the future.

## 1. Introduction

Lipomas of the cerebellopontine angle (CPA) and internal auditory canal (IAC) are relatively rare and account for approximately 0.14% of all tumors at this site [1]. On the other hand, acoustic neurinoma is the most common of all CPA tumors (85–90%) [1] and often causes hearing loss, vertigo, and tinnitus. Sometimes, acoustic neurinoma leads to brainstem compression, necessitating resection. In contrast, lipomas, which can cause symptoms similar to those of neurinomas, mostly do not require surgical treatment because their tumor growth is slower. Surgery may result in several additional deficits due to cranial nerve injuries [1]. Therefore, conservative treatment and careful observation with repeated magnetic resonance imaging (MRI) monitoring are recommended for most cases of CPA lipomas.

We report a case of simultaneous ipsilateral occurrence of CPA lipoma and acoustic neurinoma with a detailed discussion on the management and preservation of cranial nerve functions, especially the facial nerve. We also conducted a review of the literature regarding CPA/IAC lipomas.

## 2. Case Presentation

The patient was a woman in her 40s with a history of right-sided hearing loss for several years. She also had a history of acute myeloid leukemia at 14 years of age with complete remission, ovarian hypofunction, and hypothyroidism. MRI revealed a neoplastic lesion in the right CPA. Thereafter, she was referred to our department. On admission, the patient had clear consciousness and no right-sided facial nerve palsy. However, she had hearing loss with an average hearing of 23.8 dB in the right ear, assessed using the 4-frequency pure-tone average and audiogram (Figure 1). No other symptoms were found. Neurological examinations were unremarkable. MRI revealed a right CPA lesion with iso-signal intensity on T1-weighted images and heterogeneous signal intensity on T2-weighted images. Gadolinium contrast-enhanced T1-weighted imaging revealed a multicystic tumor with homogeneous enhancement. In addition, T1- and T2-weighted images revealed another tumor along the brainstem at the ipsilateral CPA, with high signal intensity and no contrast enhancement (Figure 2). On the fat suppression sequence, missing signals from the second tumor along the brainstem were observed, with no changes noted in the first tumor located in the CPA. We diagnosed these tumors as vestibular neurinomas and lipomas. The complicated anatomical position of the tumors indicated that both the facial and vestibular nerves may have adhered to the two tumors. Due to the small size of the tumor and lack of surgical experience in cases complicated by lipoma, the need for surgical treatment was decided after fully explaining this option to the patient and hearing her own preferences. The patient had already subjectively experienced hearing loss and was concerned about facial nerve palsy due to tumor expansion. Therefore, she requested early surgery. Considering that excision of lipomas predicts the risk of facial nerve palsy, we planned to selectively excise the vestibular neurinoma alone without removing the lipoma via lateral suboccipital craniotomy. The surgery was performed with continuous monitoring of the facial nerve, motor-evoked potentials, and auditory brainstem response while ensuring that the lipoma was not detached from the cranial nerve and dura. Intraoperatively, the surgical field revealed a lipoma, a yellowish cystic tumor, which covered the neurinoma (Figure 2) and part of the glossopharyngeal nerve. Care was taken not to dissect the lipoma on the facial nerve side from the seventh and eighth cranial nerves. The lipoma was partially detached from the pyramidal bone and the tumor surface to secure the extraction route, and the acoustic neurinoma was removed. Postoperative contrast-enhanced MRI indicated no residual tumor (Figure 3). There was no change in amplitude in response to intraoperative facial nerve stimulation. Postoperatively, a pathological diagnosis of neurinoma was established. Her right hearing was lost postoperatively. However, she had no other neurological deficits, including facial nerve paralysis. The patient was discharged nine days postoperatively. Written informed consent was obtained from the patient for the publication of this report.

## 3. Discussion

We performed a systematic review by retrieving the relevant literature using PubMed and Embase with the keywords “lipoma” and “schwannoma or neurinoma”. We found 151 cases of lipomas of CPA or IAC in 81 papers [2]. However, we could not find any reports on the synchronous occurrence of these two tumors.

Intracranial lipomas are rare. Approximately half of all cases of intracranial lipomas occur in the interhemispheric fissure, and CPA lipomas account for only 10% of these cases [3]. Details of the relationship between the locations of acoustic neurinoma and lipoma in the CPA or IAC, their radiological characteristics, and the typical neurological symptoms in these cases are lacking in the literature, complicating the differential diagnosis of cases with synchronous CPA lipoma and neurinoma. Additionally, symptoms of CPA lipomas, such as hearing loss (75.0%), vertigo (62.5%), and tinnitus (25.0%), were found to be quite similar to those of acoustic neurinomas [2].

In our patient, since the lipoma covered the neurinoma on its lateral side, it was necessary to mobilize the lipoma to enable resection of the neurinoma. Although the neurinoma was in contact with the lipoma, preoperative MRI and intraoperative findings suggested that the position of the lipoma was posterior and caudal to the acoustic neurinoma. These findings implied that the lipoma was on the opposite side of the facial nerve, enabling tumor removal without causing facial nerve paralysis. Subsequently, we partially separated the lipoma on the opposite side of the tumor from the pyramidal bone, where the facial nerve passed, resulting in the preservation of facial nerve function. This strategy was based on the outcomes of CPA lipoma removal in confirmed cases reported in the literature (Table 1 and Table 2) [2,3,4,5,6,7,8,9,10,11,12]. Most surgeries result in additional neurologic deficits because lipomas often adhere to the brainstem and neurovascular structures, such as the facial nerve, vestibulocochlear nerve, or other lower cranial nerves. This possibly occurs because the adipocytes of CPA/IAC lipomas might infiltrate and separate the nerve fibers, making surgical removal more difficult [13,14]. Thus, conservative treatment with repeated imaging is the most common strategy for CPA lipomas to avoid the development of additional neurological deficits. Debulking the tumor for decompression of the brainstem and cranial nerves should be considered in cases with disabling and uncontrolled neurological symptoms.

CPA lipomas are characteristically slow-growing, and mostly show no growth on follow-up. In our review, only one reported case showed an increase in size. On MRI, lipomas are hyperintense on T1-weighted images with variable intensity on T2-weighted images and do not usually enhance with gadolinium. However, acoustic neurinomas often show moderate or high enhancement.

A missing signal on the fat suppression sequence is a significant indicator of lipomas. However, no change is observed in acoustic neurinoma. These radiological features are useful for distinguishing between the two tumors.

In this case, a small acoustic neurinoma was detected in the patient and radiotherapy was considered as a treatment option. However, there were no reports on the efficacy of this treatment in patients with lipomas in close proximity. Moreover, the patient did not insist on radiotherapy, considering the side effects of radiation exposure. We conducted a literature search on intracranial lipomas using PubMed (Appendix A). However, we could not find any research reports on the efficacy of radiation therapy in treating lipomas. There was also no mention of radiation therapy in a recent systematic review of intracranial lipomas [1,2,15].

## 4. Conclusions

We report the first case of the synchronous occurrence of acoustic neurinoma and an attached CPA lipoma in the literature. The acoustic neurinoma was successfully resected while leaving the lipoma in situ, thus enabling facial nerve preservation.

## Figures and Tables

**Figure 1 diagnostics-12-00120-f001:**
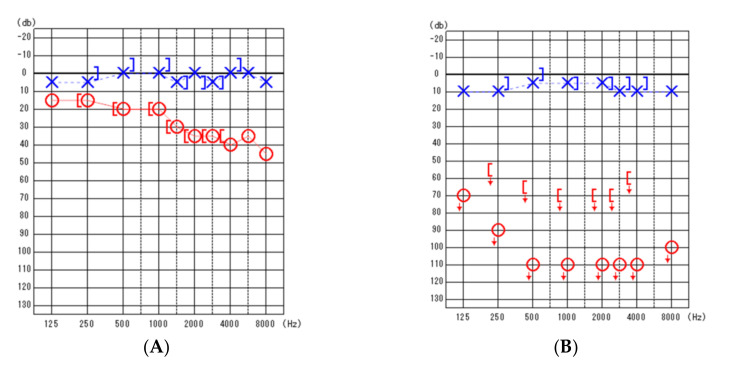
Pure-tone audiometric findings. (**A**) (**left**): Preoperative findings. (**B**) (**right**): Post-operative findings. Blue, left ear; Red, right ear; blue X, left ear air conduction threshold; red O, right ear air conduction threshold; [ , right ear measured with bone conduction threshold; ] , left ear measured with bone conduction threshold; ↓, scale out .

**Figure 2 diagnostics-12-00120-f002:**
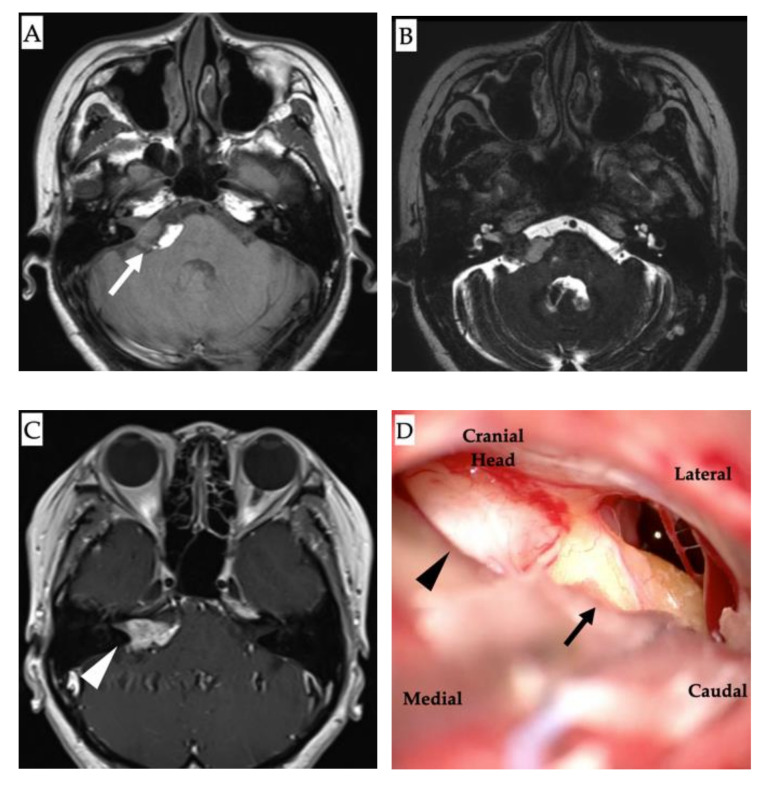
Magnetic resonance imaging (MRI) and intraoperative images. (**A**): Non-contrast axial T1-weighted MRI showing a hyperintense lesion in the right cerebellopontine angle alongside the brainstem lipoma (white arrow). (**B**): T2-weighted image showing the lipoma as a heterogeneous signal intensity lesion. (**C**): Contrast T1-weighted MRI showing a multicystic tumor with homogeneous enhancement (white arrowhead). (**D**): Intraoperative findings showing the tumors: neurinoma (black arrowhead) and lipoma (black arrow).

**Figure 3 diagnostics-12-00120-f003:**
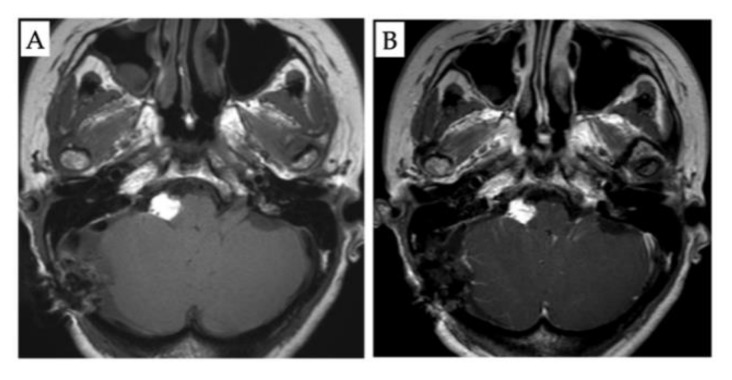
Post-operative magnetic resonance imaging (MRI) findings. (**A**): Non-contrast axial T1-weighted MRI. (**B**): Contrast T1-weighted MRI showing no residual tumor.

**Table 1 diagnostics-12-00120-t001:** Partially removed cerebellopontine angle lipomas (*n* = 24) and outcomes.

	*n*	%
No additional symptoms after operation	4	16.7
Additional symptoms after operation	20	83.3
Hearing loss	10	41.7
CN VII paresis	9	37.5
CN VI paresis	3	12.5
Vertigo	2	8.3
CN V paresis	2	8.3
CN VIII, IX, X, XI, XII paresis, CSF leakage, meningitis, cerebellar ataxia, dizziness, bitonal voice (one of each case)	10	4.2

CN, cranial nerve; CSF, cerebrospinal fluid.

**Table 2 diagnostics-12-00120-t002:** Totally removed cerebellopontine angle lipomas (*n* = 15) and outcomes.

	*n*	%
No additional symptoms after operation	1	6.7
Additional symptoms after operation	14	93.3
Hearing loss	8	53.3
CN VII paresis	7	46.7
CSF leakage	1	6.7
Severe dizziness	1	6.7

CN, cranial nerve; CSF, cerebrospinal fluid.

## Data Availability

The data presented in this study are available on request from the corresponding author.

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
