# Peer review of "Acoustic Neurinoma with Synchronous Ipsilateral Cerebellopontine Angle Lipoma: A Case Report and Review of the Literature"

_diagnostics, 2022, doi:10.3390/diagnostics12010120_

Round 1

Reviewer 1 Report

This is an interesting case report.

1. growth is rare in line 34, is slower?

2. Is the radiotherapy effect for lipoma?

Author Response

Thank you for the comments and suggestions that helped us significantly improve our manuscript. We have carefully addressed these comments and have revised the manuscript accordingly.

growth is rare in line 34, is slower?

Response: We thank you for this suggestion. Accordingly, we have corrected the phrasing as you pointed out (page: 1, line: 38).

  1. Is the radiotherapy effect for lipoma?

Response: We thank you for this query. Accordingly, we searched the literature for the effect of radiotherapy on Lipoma of CPA. However, we could not find any reports. In this case, we did not perform radiotherapy considering its adverse effects on the facial nerve. We have added these sentences to the revised manuscript. We have also added reference # [15]  (page: 5, lines: 157-161).

Reviewer 2 Report

  1. I suggest the authors should provide pre-operative and post operative pure tone audiograms and word recognition scores.
  2. I suggest the authors should provide post-operative MRI images
  3. The acoustic neureoma was small and not yet compressed brainstem. The patient still had useful hearing. The aurthors should explain why they surgically removed the tumor instead of keep observation of the tumor. The authours should explain the indications of the surgery such as tumor growing, disabling vertigo, etc.

Author Response

Thank you for the comments and suggestions that helped us significantly improve our manuscript. We have carefully addressed the comments and have revised the manuscript accordingly.

I suggest the authors should provide pre-operative and post operative pure tone audiograms and word recognition scores.

Response: We thank you for this suggestion. We have added the pre- and post- operative pure tone audiograms as you pointed out (page: 4, lines: 87).

I suggest the authors should provide post-operative MRI images

Response: We thank you for this suggestion. We have also added the post-operative MRI scans as you indicated (page: 3, lines: 78-80 and 106-113).

The acoustic neureoma was small and not yet compressed brainstem. The patient still had useful hearing. The aurthors should explain why they surgically removed the tumor instead of keep observation of the tumor. The authours should explain the indications of the surgery such as tumor growing, disabling vertigo, etc.

Response: We thank you for this suggestion. Deciding whether to surgically treat the patient was difficult due to the small size of the tumor and lack of experience in operating on patients with acoustic tumor with lipoma. We finally decided the treatment plan after listening to the patient's preference and providing sufficient information. The patient already had subjective hearing loss. The patient was also concerned about facial nerve palsy due to tumor enlargement and hoped for early surgery. These points have been added to the case presentation section  (Page: 2, Lines: 64-69).

Reviewer 3 Report

This well-written case report represents the presumably first description of a vestibular schwannoma with a synchronous ipsilateral cerebellopontine angle lipoma. By giving a brief overview, it highlights relevant information for clinicians and demonstrates key features in terms of radiologic and general diagnostic findings. Nevertheless, several minor points have to be clarified to improve the manuscript and allow the reader to fully grasp the details of every aspect described by the authors.

Minor points:

1. INTRODUCTION

Brief and adequate overview of important factors affecting the topics of interest.

- However, “acoustic schwannoma” is a rather uncommon term. Usually, either “acoustic neuroma/neurinoma” or “vestibular schwannoma” are preferred.

2. CASE PRESENTATION

Overall, clear presentation of the case.

- I have never heard of the “4-min method”, please specify (citation)? Is it pure tone average for four different frequencies? The latter is more commonly used, so a description of the patient’s hearing with it is probably easier to understand for most readers.

- Furthermore, the authors write that “facial nerve sensitivity was not affected during surgery”. Do you mean that facial nerve monitoring demonstrated an intact nerve?

3. DISCUSSION

Appropriate discussion of the topic.

- While most neurological deficits are clear, I would like to know more about the described CN VIII paresis as hearing loss and vertigo are already mentioned separately in the table?

4. CONCLUSION

OK.

AUTHOR CONTRIBUTIONS

There are still several locations in the text where X.X. has been left instead of adding the authors’ initials.

Author Response

Thank you for the comments and suggestions that helped us significantly improve our manuscript. We have carefully addressed the comments and have revised the manuscript accordingly.

This well-written case report represents the presumably first description of a vestibular schwannoma with a synchronous ipsilateral cerebellopontine angle lipoma. By giving a brief overview, it highlights relevant information for clinicians and demonstrates key features in terms of radiologic and general diagnostic findings. Nevertheless, several minor points have to be clarified to improve the manuscript and allow the reader to fully grasp the details of every aspect described by the authors.

- However, “acoustic schwannoma” is a rather uncommon term. Usually, either “acoustic neuroma/neurinoma” or “vestibular schwannoma” are preferred.

Response: We thank you for this suggestion. As you suggested, I have unified the terminology into “acoustic neurinoma” throughout the manuscript including: the title abstract, and key words.

- I have never heard of the “4-min method”, please specify (citation)? Is it pure tone average for four different frequencies? The latter is more commonly used, so a description of the patient’s hearing with it is probably easier to understand for most readers.

Response: We thank you for this comment. The English description was incorrect and has been corrected into “ the 4-frequency pure-tone average” (page: 2, line: 54).

- Furthermore, the authors write that “facial nerve sensitivity was not affected during surgery”. Do you mean that facial nerve monitoring demonstrated an intact nerve?

Response: We thank you for this query. The English description was incorrect and has been corrected.

“There was no change in the amplitude to the intraoperative facial nerve stimulation”. (Page: 2, line: 79)

- While most neurological deficits are clear, I would like to know more about the described CN VIII paresis as hearing loss and vertigo are already mentioned separately in the table?

Thank you for your insightful remarks.

There is a report describing the clinical course of 84 cases. Of these, 64 had hearing impairment, and 19 had postoperative hearing loss.

There are few reports that describe details of hearing impairment, and the number of references is markedly increased. We have created an Excel file including the results of our survey as supplemental table.

This manuscript is a resubmission of an earlier submission. The following is a list of the peer review reports and author responses from that submission.